# Implementation of Disassembler on Microcontroller Using Side-Channel Power Consumption Leakage

**DOI:** 10.3390/s22155900

**Published:** 2022-08-07

**Authors:** Daehyeon Bae, Jaecheol Ha

**Affiliations:** 1School of Cybersecurity, Korea University, Seoul 02841, Korea; 2Division of Computer Engineering, Hoseo University, Asan 31499, Korea

**Keywords:** Internet of Things, hardware security, side-channel analysis, side-channel-based disassembler, machine learning

## Abstract

With the development of 5G and network technology, the usage of IoT devices has become popular. Because most of these IoT devices can be controlled by an adversary away from the administrator, several security issues such as firmware dumping can arise. Firmware dumping is the cornerstone or goal of many types of hardware hacking. Therefore, many IoT device manufacturers adopt some protection mechanisms such as the restriction of hardware debuggers. However, several recent studies have shown that the operating instructions of an IoT device can be recovered through the profiling-based side-channel analysis. The Side-Channel-Based Disassembler (SCBD) refers to software that recovers instructions of the device only from the side-channel signal. The SCBD is powerful enough to defeat many firmware protection mechanisms. In this paper, we show how an adversary can build an instruction (opcode)-level disassembler using the power consumption signal of commercial microcontrollers (MCUs) such as the 8-bit ATxmega128 and 32-bit STM32F0. To implement the SCBD, we elaborately constructed the instruction template considering the pipeline of the target MCUs through instruction sequence analysis. Furthermore, we preprocessed the side-channel signals using the Continuous Wavelet Transform (CWT) for noise reduction and Kullback-Leibler Divergence (KLD) for instruction feature extraction. Our experimental results show that the machine-learning-based instruction disassembling models can recover the operating instructions with an accuracy of about 91.9% and 98.6% for the ATxmega128 and STM32F0, respectively. Furthermore, we achieved an accuracy of 77% and 96.5% in a cross-board validation.

## 1. Introduction

With the development of 5G and network technology, numerous IoT devices are connected to the network. These IoT devices often play an important role and deal with sensitive information. However, many IoT devices equipped with commercial microcontrollers operate in an environment that anyone can physically access and control. A typical security threat that can occur in such an environment is firmware dumping, which is the cornerstone or goal of hardware hacking and can cause problems such as infringement of intellectual property rights and exposure of private security mechanisms. Therefore, most IoT device manufacturers adopt firmware protection mechanisms and restrict access from the hardware debuggers to thwart reverse engineering. However, recent research showed that the instructions operated in the microcontrollers can be recovered through profiling-based side-channel analysis. That is, only the signal leaked from the hardware can be used to recover the instruction executed inside the processor. The SCBD is powerful enough to defeat many firmware protection mechanisms such as the restriction of the hardware debugger.

In this paper, we implement the opcode-level SCBD of 8-bit and 32-bit commercial microcontrollers using the power consumption signal. Specifically, we implement an SCBD for the Atmel ATxmega128 and STMicroelectronics STM32F0, which are embedded in numerous commercial low-end IoT devices. To achieve this goal, we elaborately constructed the instruction template considering the pipeline of the target MCUs. Furthermore, we preprocessed the side-channel signal using the CWT for noise reduction and the KLD for the instruction feature extraction. In this phase, we introduce the instruction sequence analysis, the scale optimization of the mother wavelet, and the k-fold feature extraction to improve the recovery accuracy. Our experimental results show that the machine-learning- and deep-learning-based instruction disassembling models can recover the operating instructions with high accuracy. We achieved a recovery accuracy of 91.9% and 77.0% for the model validation data and cross-board data of the ATxmega128, respectively. In addition, in the case of the STM32F0, we show that the data for model validation and the cross-board data can be recovered with 98.6% and 96.5% accuracy, respectively. We represent the flow of our SCBD from dataset construction to the model application phase in Figure 1.

**Contributions:** We implemented the SCBD for the 8-bit ATxmega128 and the 32-bit STM32F0, which are commercial microcontrollers, using the side-channel power consumption leakage. We finally designed the deep learning model as an SCBD adopting various normalization methods: dropout, batch normalization, and residual blocks. In addition, we performed the cross-board accuracy test considering the actual operating environment of the SCBD (profiling scenario). As a result, we confirmed that our SCBD targeting a 32-bit processor (STM32F0) can outperform previous results in terms of cross-board accuracy.

## 2. Background

### 2.1. Side-Channel Analysis and Power Analysis Attack

Side-channel analysis, proposed as the timing attack by P. Kocher in 1996, is a type of passive attack that extracts secret information in the device by analyzing side-channel signals leaking from the hardware [1]. Here, the side-channel signal refers to all kinds of signals that are unintentionally leaked through illegitimate channels from the hardware. There are some representative examples of the side-channel signal such as electromagnetic radiation, power consumption, heat, sound, operating timing, and so on. Side-channel analysis has been mainly studied in the field of cryptanalysis to recover the secret key embedded in the cryptographic device and to design countermeasures against it. In addition, early side-channel analysis research was conducted targeting smart cards and microprocessors having a simple hardware architecture. However, some research shows that commercial computer processors, as well as smartphones can be vulnerable to side-channel analysis. Therefore, research on high-performance devices has been actively conducted in recent years [2,3,4,5].

The power analysis attack, the most active research field of side-channel analysis, is an analysis method using the power consumption signal of a device. Examples of the power analysis attack include differential power analysis (DPA) [6], correlation power analysis (CPA) [7], and differential deep learning analysis (DDLA) [8,9]. The power analysis attack is performed based on the fact that the side-channel signal is dependent on the instruction or data processed on the device, and these two dependencies are called operation dependency and data dependency, respectively. For example, the total power consumption of the microcontroller $P_{total}$ can be described by Equation (1). Here, $P_{op}$ is the power consumption by the operation dependency, $P_{data}$ is the power consumption by the data dependency, $P_{e\_noise}$ is the electrical noise, and $P_{const}$ is the constant power consumption factor [10].
(1)$$P_{total}=P_{op}+P_{data}+P_{e\_noise}+P_{const}$$

In our simple experiments (using the ChipWhisperer CW308T-XMEGA), the degree of influence on power consumption is large in the order of $P_{op}>P_{data}>P_{e\_noise}$, and the electrical noise has a normal distribution, i.e., $P_{e\_noise}\sim N\left(0,\sigma \right)$. In fact, as a result of the experiment using an 8-bit microcontroller ATxmega128, we confirmed that $P_{data}$ is $P_{op}/3$ and $P_{e\_noise}$ is $P_{op}/30$, approximately. The specific degree of influence will be different for each type of microcontroller, but it is common to have a large degree of influence in the order of $P_{op}>P_{data}>P_{e\_noise}$. In addition, the power consumption includes only the signal for the entire microcontroller, not the signal of a specific component of the microcontroller due to the limitation of the measurement environment. Note that microcontrollers produced in different processes are likely to have different characteristics. Most microcontrollers use pipeline technology to execute instructions in parallel, so the power is consumed by behavior that is not related to the instruction or data to be analyzed. The power consumption due to this behavior is an obstacle to side-channel analysis, so it can be regarded as noise, which is called switching noise. 

Side-channel analysis targeting the cryptographic algorithm can be performed using $P_{data}$ or $P_{op}$. In the power analysis attack targeting $P_{data}$, a model that can calculate the predicted power consumption according to the processed value is required. Due to the limited information of the microcontroller, the Hamming distance and the Hamming weight models, which can calculate the relative difference in the predicted power consumption using only the data values, are mainly used. In the power analysis attack targeting $P_{op}$, timing information or data-dependent information can be utilized in SPA, single-trace attack (using correlation coefficient), and so on. To protect cryptographic algorithms from side-channel analysis, software-based [11] and hardware-based [12] countermeasures have been continuously developed. These countermeasures based on their specific design protect not only $\;P_{data}$, but also $P_{op}$ from the advanced side-channel analysis attacks including DPA, CPA, etc.

The Hamming distance is a power model based on the characteristics of a complementary metal–oxide-semiconductor (CMOS). It is based on the assumption that $P_{data}$ is proportional to the number of flipped bits because the power is consumed in the process of flipping bits of data. On the other hand, the Hamming weight model is based on the assumption that the power is consumed proportional to the number of “1”s in the bit string because the bus is pre-charged. Using this power model, the predicted power consumption for the intermediate operation value of the cryptographic algorithm can be calculated, and the secret key can be recovered by analyzing it with the side-channel signal. In fact, the leakage mechanism of the processor is too complex to model, but data-dependent attacks targeting cryptographic algorithms can adopt a simple data-driven power model. Our disassembler needs to use much information such as instructions and data. In this case, the modeling process for the power signal can be omitted by using deep learning techniques (analysis of standard deviation caused by random instruction/data).

### 2.2. Side-Channel-Based Disassembler

Side-channel analysis can also be used to develop an instruction disassembler rather than cryptanalysis, and software that recovers the instruction from a side-channel signal is called a side-channel-based disassembler. This is premised on the fact that there is a slight difference in the power consumption because when the different machine code is fetched, decoded, and executed, different types of logic circuits are operating. This means that the instruction executed inside the microcontrollers can be recovered using $P_{op}$, which occupies the largest portion in the above Equation (1).

As a research case of the SCBD, the power consumption and the electromagnetic radiation were used as side-channel signals, and attempts have been made at the machine code level and the opcode level as a recovery unit. The machine-code-level recovery treats the instruction as only a bit string and recovers the encoded code. Therefore, the instruction template building phase can be performed only by generating a random bit string, so it is not dependent on a specific microcontroller and is flexible. In addition, since machine code includes all operands such as literal constants and the registers, it can also be recovered without additional work. However, much effort is required in the signal acquisition phase because a side-channel signal in the instruction fetch process with small switching noise is required. On the other hand, the opcode-level SCBD recovers the identifier assigned according to the mnemonic. For this, much effort for template building is required because it is necessary to compose an assembly language that conforms to the grammar of a specific microcontroller. In addition, a model for recovering operands, e.g., the registers and literal constants, is additionally required for each instruction (opcode). However, there is an advantage that recovery is possible even if a power consumption signal with much switching noise is used. 

A study on the SCBD was first attempted by Vermoen et al. in 2007 [13]. Vermoen et al. attempted power consumption signal analysis on Java cards and showed that they could recognize 10 different Java bytecodes with at least 90% accuracy. They correlated the power consumption signal with the average power template of each byte code and classified it as a byte code. However, they did not propose a specific instruction recovery algorithm. Eisenbarth et al. implemented an instruction classifier by performing a template attack on a PIC16F687 microcontroller using the power consumption signal [14]. They achieved a recognition rate of 70.1% in 35 test instructions by applying statistical models such as the hidden Markov model (HMM). Strobel et al. implemented an instruction-level disassembler for the PIC16F687 microcontroller using multiple electromagnetic channels [15]. They showed a recognition rate of 96.24% in the test instruction without the usage of a statistical model such as the HMM. Park et al. showed that 112 instruction codes and 64 registers can be recovered with an accuracy of 99.0% for the ATMega328P microcontroller using signal preprocessing techniques such as the CWT, principal component analysis, and the classification model proposed by McCann et al. [16,17]. Recently, in 2019, V. Cristiani et al. implemented a disassembler through a machine-code-level (bit-level) approach for the first time, showing a new development potential that is more flexible and can fully recover all operands, including literal constants [18]. However, there is a difficulty in that it is necessary to collect a local side-channel signal leaking only during the instruction fetching process.

### 2.3. Specification of Microcontrollers

In this subsection, we discuss the two types of microcontrollers that were analyzed: Atmel ATxmega128 and STMicroelectronics STM32F0.

#### 2.3.1. Atmel ATxmaga128

The ATxmega128 is an 8-bit unit Atmel microcontroller with a low-power, high-performance processor of the AVR Reduced Instruction Set Computer (RISC) architecture. The ATxmega128(D4) is equipped with 128 KB of flash memory and 8 KB of SRAM for program code storage and can operate with a voltage of 1.6~3.6 V and a clock cycle of up to 32 MHz. It adopts the Harvard architecture in which data memory and instruction memory are separated and has an independent program bus and data bus so that the instruction and data can be accessed at the same time. Additionally, 32 8-bit unit general-purpose registers R0~R31 are provided, of which R26~R31 can be used as pointer registers, which can store 16-bit addresses. The pointer register supports three types of X, Y, and Z, and is used as the concatenated value of the R27~R26, R29~R28, and R31~R30 registers, respectively. In addition, since a single-level pipeline with two stages is adopted, when an instruction is executed in one clock cycle, the next instruction is fetched simultaneously, as shown in Figure 2.

The power consumption signal of the ATxmega128, as well as other microcontrollers shows a very similar pattern for each clock cycle, only the distribution of the value is slightly different. In fact, Figure 3 shows the power consumption trace for each clock cycle when the ATxmega128 microcontroller operates at 4MHz. Here, Channel 1 (yellow) is the power consumption and Channel 4 (green) is a clock signal. Most power is consumed at the rising edge, and less power is consumed at the falling edge.

#### 2.3.2. STMicroelectronics STM32F0

The STM32F0 is a microcontroller of STMicroelectronics based on a 32-bit Cortex-M0 processor of the ARM architecture. The STM32F0 is equipped with 128 KB of flash memory for program code and 16KB of SRAM and can operate with a voltage of 2.0~3.6V and a clock cycle of up to 48 MHz. The Cortex-M0 is an ARMv6-M architecture and has a structure in which a data bus and an instruction bus are integrated by adopting the von Neumann architecture. In addition, the instruction is executed through a three-stage pipeline consisting of fetch, decode, and execution, as shown in Figure 4. The Cortex-M0 provides 13 general-purpose registers and special registers of 32-bit size. The registers from R0 to R12 are provided as the general-purpose registers, of which R0 to R7 are called “low registers” and R8 to R12 are called “high registers”. Some instructions, such as MOV, can use any register as an operand, but many Thumb instructions can only use lower registers due to the space limitation of the 16-bit Thumb instruction. In addition, the stack pointer, the link register, and the program counter are allocated to R13, R14, and R15, respectively. 

## 3. Instruction Template Building and Power Consumption Signal Acquisition

As seen in Section 2, the power consumption signals of the instruction are not intuitively distinguishable; only the distribution of the values is slightly different. Therefore, most SCBDs rely on machine learning, which requires a large amount of dataset. This section describes how to build an instruction template.

### 3.1. Instruction Template Building

All stages of the pipeline are executed for each clock cycle in parallel, so the components of multiple instructions are contained in the power consumption trace of one clock. For this reason, the range of effect that one instruction can affect the execution process of another instruction is determined by the number of stages in the pipeline. Therefore, to create the instruction template, the number of pipeline stages of the microcontroller should be firstly considered. For example, the ATxmega128 has two stages of fetch and execution, which means that, while one instruction is being executed, the next instruction is fetched. That is, the power consumption signal corresponding to each clock cycle includes one instruction execution and the next instruction fetch power signal. Furthermore, when the operating frequency of the microcontroller increases, the power consumption signal may overlap because the next instruction is executed before the capacitor is fully recharged. Therefore, to implement the SCBD, the template must be configured in units of the number of stages in the pipeline + 1 instructions. For example, Listing 1 shows a part of the C language source code of the firmware in which the target instruction is inserted as an inline assembly. In this case, the target instruction is the ADC in Line 18. On the other hand, in the case of STM32F0, since the pipeline consists of three stages, the length of the instruction template excluding the NOP should be four and the target instruction should be in the second place.
**Listing 1.** A firmware source code used in our experiments (written in C language).1.unsigned char execute()2.{3.  PUSH_REGS;          ▷ Backup registers4.  trigger_high();5.  asm volatile(           ▷ In-line assembly for instruction template6.    “NOP \n\t”7.    “ADD r3, r17 \n\t”8.    “ADC r26, r7 \n\t”      ▷ Target instruction9.    “MOV r10, r11 \n\t”10.    “NOP \n\t”11.  );12.  trigger_low();13.  POP_REGS;            ▷ Restore registers14.  Return 0;15.}

### 3.2. Instruction Sequence Analysis and Selection

In most SCBD studies, the opcodes and operands are randomly selected from a uniform distribution when composing the instruction template, as shown in Listing 1 above. However, if the opcode is selected from the uniform distribution to build the instruction template, some problems may occur in the real-world instruction recovery. In other words, it will show a high recovery rate for the dataset constructed from the template, like the above Listing 1, but it will show poor performance for other firmware (not part of the template). Therefore, only instruction sequences that can actually be executed in hardware should be constructed as templates. To select instructions that can actually operate on the hardware, we compiled and analyzed real block cipher algorithms instead of using the hidden Markov model (HMM). This method can be simply used when extracting instructions as an alternative to the complex HMM. Here, the instruction sequence means a list of N-instructions that are continuously executed, and N can be determined as the number of pipeline stages +1 considering the signal overlap and the pipeline.

The compiler does not randomly combine the instructions, but often uses a combination of the instructions corresponding to a specific syntax. That is, there is a very high probability that the comparison or value increase/decrease operator will appear immediately before the branch instruction, and the operation of the cryptographic algorithm is often repeated due to the small word size of the microcontroller, e.g., the repetition of data load instructions. For this reason, to recover the real-world instructions generated by the compiler, it will be advantageous to learn the instruction combinations that appear with a high frequency. Therefore, we performed instruction sequence analysis. 

In this paper, we analyzed the instruction sequence of several block cipher algorithms, for example. The block cipher algorithms analyzed were AES-128 [19], ARIA-128 [20], SEED-128 [21], and LEA-128 [22], all of which use firmware generated by compiling standard codes. A total of 20 firmware were created by applying all possible optimization options −O0~−O3, and −Os, and the frequency of the instruction sequence was analyzed. As a result of the analysis, we confirmed that the instructions (opcodes) generated by AVR-GCC consist of only a few patterns. We show the result of the “ADC” opcode analysis in Figure 5.

Theoretically, if the ADC instruction is fixed and the previous and following instructions are randomly selected, there can be $137^{2}$ (about 18 thousand) combinations. However, the compiler generates only 50 instruction sequences among them. Therefore, if the template is composed of frequently used instruction sequences, better performance can be achieved in real-world instruction recovery. As a result of analyzing other opcodes of the ATxmega128 in the same way, we confirmed that the compiler uses at least 5 and, at most, about 200 sequences in our experimental environment. In addition, we confirmed that the ratio of the top 30 sequences for all opcodes occupies 80% or more in each opcode. Therefore, if the template is composed of the top 30 sequences, a higher recovery rate can be seen in the real-world code. 

In the same way, the instruction sequence of the STM32F0 should be analyzed. Since the STM32F0 needs to be analyzed in units of four instructions, more types of sequences than the ATxmega128 were used. However, we also confirmed that some STM32F0 instruction sequences are used repeatedly. The proportion of the top 30 instruction sequences varied from 20% to 100%, but half of them occupy 100%. Therefore, we created the instruction templates for the top 30 instruction sequences of the ATxmega128 and the STM32F0. We did not include opcodes that are not used at all or used five times or less in the template. The supported instruction sets and the target opcodes for the ATxmega128 and the STM32F0 are shown in Table 1.

### 3.3. Acquisition of Instruction Power Consumption Signal

In this paper, we used ChipWhisperer [23] target boards and a LeCroy oscilloscope to create a power consumption template for the instructions. Specifically, the CW308T-XMEGA target board equipped with the ATxmega128 and the CW308T-STM32F target board equipped with the STM32F0 were used for the instruction template building. Additionally, there was no clock generator on the target board. Therefore, we connected the ChipWhisperer-Lite capture board to supply the clock and relay serial communication with the PC. Since the ChipWhisperer capture board supports the Atmel PDI and STM serial boot loader, we deployed firmware in a unified way using only the capture board without the hardware debugger. 

To measure the power consumption signal of the instructions, we constructed an experimental environment with the structure shown in Figure 6. The oscilloscope used in the experiment was a LeCroy HDO4021, which supports a signal bandwidth of up to 200 MHz, 12-bit resolution, and a sampling rate of 10 GS/s. In this experiment, the signal was measured at 200 MHz without limiting the bandwidth, the resolution was set to 8 bit, and the sampling rate was set to 1.25 GS/s. Furthermore, the same 4 MHz clock was supplied to the two target boards. In this measurement environment, we measured the power consumption trace of the instructions, as shown in Table 2. In addition, we separated the target board and the test board to measure the traces for a real-world scenario. Here, we refer to the test board as the cross-board.

## 4. Signal Preprocessing and Instruction Feature Extraction

The power consumption signal measured with an oscilloscope not only contains electrical noise, but also the signal alignment is not perfect. Therefore, we transformed the time-domain signal into the time–frequency-domain data with the CWT for noise reduction and signal alignment. We extracted the feature points of the instructions by utilizing the KLD to reduce the computational complexity required to implement the disassembler and improve the recovery accuracy.

### 4.1. Signal Preprocessing Using Continuous Wavelet Transform

#### 4.1.1. Continuous Wavelet Transform

The wavelet transform is one of the multi-resolution time–frequency analysis techniques, and it decomposes a signal in a one-dimensional time domain into data in a two-dimensional time–frequency domain. This method can overcome the disadvantage that the short-time Fourier transform (SFTF) has a trade-off between the frequency resolution and the time resolution due to the uncertainty principle. The wavelet transform function F is described by Equation (2). Here, the continuous function of the signal to be analyzed with time t is $x\left(t\right)$, and the mother wavelet is $\psi \left(t\right)$; the scale factor is s, and the transition factor is τ.
(2)$$F_{\psi }\left(\tau ,s\right)=\frac{1}{\sqrt{\left|s\right|}}\int_{-\infty }^{\infty }x\left(t\right)\psi \left(\frac{t-\tau }{s}\right)dt$$

The mother wavelet ψ must be the sum of all areas equal to 0, as described by Equation (3), and is a function of a short oscillation form that increases and decreases based on 0. The mother wavelet can be expanded or compressed according to the frequency to be extracted, which is determined by the scale factor s. Examples of well-known wavelets include Mexican hat, Morlet, Gaussian, and Daubechies, and all experiments in this paper used Mexican hat wavelets.
(3)$$\int_{-\infty }^{\infty }\psi \left(t\right)dt=0$$

The CWT performs a convolution operation with the target signal and the mother wavelet rescaled according to s. In this phase, the effect of noise reduction can be seen because the electrical noise that follows the Gaussian distribution and exists in all samples is accumulated and canceled when calculating the low-frequency signal. In addition, since the convolution operation is performed on many samples and combined into one CWT coefficient, it is insensitive to alignment.

#### 4.1.2. Optimized Scale Selection of Mother Wavelet

In the CWT, the frequency of the signal that can be extracted is determined according to the scale of the mother wavelet. That is, the relationship between the scale of the mother wavelet and the extracted frequency is as described by Equation (4). Here, $f_{s}$ is the frequency to be extracted, $f_{m}$ is the center frequency of the mother wavelet, and Δ is the sampling period.
(4)$$f_{s}=\frac{f_{m}}{\Delta \times s}$$

The scale s of the mother wavelet is an important parameter that must be determined when applying the CWT. However, in most related studies, there is no clear standard, such as selecting the minimum and maximum scales as random natural numbers and using continuous natural numbers within the corresponding range as the scale. In the SCBD implementations, however, the scale of the mother wavelet must be carefully determined as it can affect whether the real-world code can be recovered. That is, if the maximum scale of the mother wavelet is not limited, a fixed instruction component (e.g., NOP, trigger-related instructions) existing only in the template may be included. Therefore, the maximum scale should be limited, as described by Equation (5), for the SCBD implementation. Here, c is the maximum number of instructions to be combined, and $f_{r}$ is the operating clock frequency.
(5)$$s=\frac{f_{m}}{\Delta \times \frac{f_{r}}{c}}$$

In our experiment environment, c is 2 because a signal of 2 clocks is used, $f_{m}$ is 0.25, $f_{r}$ is $4\times 10^{6}$ Hz, and Δ is $1/\left(1.25\times 10^{9}\right)$, so the maximum scale s is determined to be 156.25. In fact, we do not know which frequency band contains meaningful information. Therefore, we used the value obtained by dividing the space between 0 and 156.25 into 100 equal parts as the scale. As a result, only meaningful features can be automatically selected by the deep learning model. 

### 4.2. Feature Extraction Using Kullback-Leibler Divergence

#### 4.2.1. Kullback-Leibler Divergence 

The KLD is a function that calculates the difference between two given probability distributions P and Q. Since the KLD refers to the difference in the information entropy that occurs when sampling using a Q distribution that approximates the P distribution, it is called the relative entropy. The information entropy of the discrete probability distribution P can be calculated as described by Equation (6). The information entropy when sampling with a Q distribution that approximates P can be described by Equation (7). Therefore, the information entropy difference between the two probability distributions P and Q can be described by Equation (8).
(6)$$-\underset{i}{\sum }P\left(i\right)\log_{2}P\left(i\right)$$
(7)$$-\underset{i}{\sum }P\left(i\right)\log_{2}Q\left(i\right)$$
(8)$$-\underset{i}{\sum }P\left(i\right)\log_{2}Q\left(i\right)-(-\underset{i}{\sum }P\left(i\right)\log_{2}P\left(i\right))$$

The preceding Equation (8) can be briefly expressed as the following Equation (9). Therefore, the KLD for two discrete probability distributions P and Q can be described by Equation (9).
(9)$$\underset{i}{\sum }P\left(i\right)\log_{2}\frac{P\left(i\right)}{Q\left(i\right)}$$

The KLD is not a distance function because the result changes when the order of two given probability distributions P and Q is changed. If the value of the KLD is 0, it means that the two probability distributions are equal. Furthermore, when it is greater than 0, it has a value proportional to the difference between the two probability distributions.

#### 4.2.2. Feature Extraction Using KLD

The power consumption signal of instructions may have a different distribution of values even for the same instruction due to the influence of the pipelines, the signal overlap, the operands, the operation values, etc. Therefore, only samples that satisfy the following two characteristics (1) and (2) can be regarded as the characteristics of the instruction:

The power consumption should have a similar distribution for the same instructions (opcodes).The power consumption should have a different distribution for the other instructions (opcodes).

Therefore, to extract the feature points that satisfy the two characteristics, the feature extraction algorithms shown in Algorithms 1 and 2 were used. Following Algorithm 1 is an algorithm for extracting the unique feature points that satisfy Feature (1), and Algorithm 2 is an algorithm for extracting distinct feature points that satisfy Feature (2). Finally, the points extracted simultaneously in both algorithms are considered as the features of the instruction. Here, the foldnum represents the number of repetitions for feature extraction to improve reliability. The divider is used to calculate the indices of power traces.
**Algorithm 1** Extraction of the feature points representing the unique characteristics of each instruction INPUT: $Trace_{CWT}$, INST, Divider d, Foldnum fd, KLD threshold $TH_{K}$, Foldnum threshold $TH_{F}$
 OUTPUT: Unique feature points $P_{unique}$
1.**function** calc_KLD($group_{a},\;group_{b}$)2.    Calculate and return the KLD values for all points between $group_{a}$ and $\;group_{b}$
3.**function** extract_unique_points($KLD_{ab},\;\;TH_{K},\;\;TH_{F}$)4.    Extract and return all points that pass threshold-based tests5.    ▷ The KLD must be less than $TH_{K}$
6.    ▷ The points must be extracted at least $TH_{F}$ times for reliability7.
8.$P_{unique}\leftarrow$ []9.**for each** $i_{target}\in Inst$ **do**10.    $KLD_{ab}\leftarrow$ []11.    shuffle($Trace_{CWT}\left[i_{target}\right]$)12.    N ← length($Trace_{CWT}\left[i_{target}\right]$))13.    $group_{a}$ $\leftarrow Trace_{CWT}\left[i_{target}\right]\left[0\ldots N/d\right]$14.    $group_{b}$ $\leftarrow Trace_{CWT}\left[i_{target}\right]\left[N/d\ldots 2\times N/d\right]$15.**    for** fold=1 **to**fd **do**16.$\;KLD_{ab}\left[fold\right]\leftarrow calc\_KLD\left(group_{a},\;group_{b}\right)$17.    $P_{unique}\left[i_{target}\right]\;\leftarrow$ extract_unique_points($KLD_{ab}$, $TH_{K}$, $TH_{F}$))18.**return** $P_{unique}$

**Algorithm 2** Extraction of the feature points representing the distinct characteristics of each instruction INPUT: $Trace_{CWT}$, INST, Divider d, Foldnum fd, KLD threshold $TH_{K}$, Foldnum threshold $TH_{F}$
 OUTPUT: Unique feature points $P_{distinct}$
1.**function** calc_KLD($group_{a},\;group_{b}$)2.    Calculate and return the KLD for all points between $group_{a}$ and $group_{b}$
3.**function** extract_distinct_points($KLD_{ab},\;TH_{K},\;TH_{F}$)4.    Extract and return all points that pass threshold-based tests5.    ▷ The KLD must be greater than $TH_{K}$
6.    ▷ The points must be extracted at least $TH_{F}$ times for reliability7.
8.$P_{distinct}\leftarrow$ []9.
**for**
**each**

$i_{target}\in Inst$

**do**
10.
**    for each**

$i_{2}\in Inst$

**do**
11.**        if**$i_{target}$ == $i_{2}$
**do**12.
**            continue**
13.        $KLD_{ab}\leftarrow$ []14.        shuffle($Trace_{CWT}\left[i_{target}\right],\;Trace_{CWT}\left[i_{2}\right]$)15.        N ← min(length($Trace_{CWT}\left[i_{target}\right]$)), length($Trace_{CWT}\left[i_{2}\right]$)))16.

$\;\;\;\;\;\;group_{a}\;\leftarrow Trace_{CWT}\left[i_{target}\right]\left[0\ldots N/d\right]$

17.

$\;\;\;\;\;\;group_{b}\;\leftarrow Trace_{CWT}\left[i_{2}\right]\left[0\ldots N/d\right]$

18.        **for**
fold=1
**to**
fd
**do**19.            $KLD_{ab}\left[fold\right]\leftarrow calc\_KLD\left(group_{a},\;group_{b}\right)$
20.        $P_{distinct}\left[i_{target}\right]\left[i_{2}\right]\;\leftarrow$ extract_distinct_points($KLD_{ab}$, $TH_{K}$, $TH_{F}$))21.
**return**

$P_{distinct}$



We extracted 4696 instruction feature points of the ATxmega128 and 2343 feature points of the STM32F0 using the above two algorithms. For example, the following Figure 7 shows 4696 feature points of the ATxmega128. The dataset corresponding to the extracted feature points is used to train the machine learning and the deep learning models in Section 5.

## 5. Implementation of the Instruction Disassembler

### 5.1. Instruction Recovery Using the Machine Learning and Deep Learning Models

We implemented machine-learning- and deep-learning-based instruction disassemblers using the final feature points extracted in the previous Section 4.1. To implement the SBCD, K-nearest neighbor (KNN) and random forest (RF) as the machine learning models were used, and the convolutional neural network (CNN) and the multi-layer perceptron (MLP) as the deep learning models were used.

#### 5.1.1. Instruction Recovery Using KNN

KNN is one of the simplest machine learning algorithms, and the learning phase is simply to express training data in the coordinates. In the inference phase of new data, after expressing the given data in the existing coordinates, the label is determined based on the K nearest neighbors. 

In the case of KNN, the training and the validation phases are performed based on the extracted features without performing the data preprocessing due to the characteristics of KNN. The hyperparameter k, which is the number of nearest neighbors, is 3 in both the ATxmega128 and the STM32F0. As a result of the ATxmega128 instruction classification, the recovery accuracy of the validation data is 90.8% and the cross-board data accuracy is 64.2%. In addition, in the case of STM32F0, we confirmed that the validation data can be recovered with an accuracy of 98.8% and cross-board data of 84.6%.

#### 5.1.2. Instruction Recovery Using RF

RF is a kind of ensemble model composed of multiple decision trees. The performance of the decision trees varies greatly depending on the randomness of the training data. RF can improve classification performance by using several slightly different decision trees. Therefore, RF uses some methods such as bootstrap and bagging to provide randomness to each decision tree. 

For the training and recovery using the RF model, the mean of the input data is adjusted to 0 and the standard deviation is adjusted to 1. The number of internal trees, which is a hyperparameter of RF, uses 1500 for both the ATxmega128 and the STM32F0. As a result of the ATxmega128 instruction classification, the recovery accuracy of the validation data is 91.3% and the cross-board data accuracy is 59.7%. In addition, in the case of the STM32F0, we confirmed that the validation data can be recovered with an accuracy of 98.7% and cross-board data of 81.6%.

#### 5.1.3. Instruction Recovery Using CNN

The CNN is the most popular deep learning model for its excellent performance in image recognition and speech recognition. It consists of convolutional layers for feature extraction and a multi-layer perceptron for classification. In the convolution layer, the convolution operations are performed on the input data and the filter (so-called kernel or feature map), so that the correlation with the neighboring data can be reflected. 

To classify instructions using the CNN, we selected one-dimensional CNN rather than two-dimensional CNN. In general, the two-dimensional CNN performs training while maintaining association with neighboring data such as the pixels of a picture. Since the data used in our experiment were extracted from the spectrogram and concatenated, as shown in Figure 7, there is no correlation between neighboring data. Considering this point, we adopted one-dimensional CNN in this experiment.

In the CNN model, the multi-layer perceptron for classification consists of three hidden layers with 2048, 1024, and 512 nodes, respectively. As in RF, the mean of the input data was adjusted to 0 and the standard deviation was adjusted to 1. In addition, the importance of features was analyzed using RF, and only the top 1500 important features were extracted and used. As a result of ATxmega128 instruction classification, the recovery accuracy of validation data is 66.7% and the cross-board data accuracy is 54.4%. In addition, in the case of STM32F0, we confirmed that the validation data can be recovered with an accuracy of 84.8% and cross-board data of 74.2%. We speculate that the reason why the CNN model performs poorly is that the data used in the experiment already preceded the feature extraction, i.e., there is no correlation with the contiguous samples.

#### 5.1.4. Instruction Recovery Using MLP

The perceptron is a simple algorithm that accumulates multiple inputs, applies an activation function, and returns it. The MLP, a so-called neural network, is a hierarchical stack of these perceptrons and consists of an input layer, hidden layers, and an output layer. In addition, it is also called a feed-forward network because all values are passed from the input layer to the output layer. The training phase of the MLP is to approximate a function by updating the weights.

First, we tried to recover using an MLP with three hidden layers consisting of 1024, 512, and 256 nodes, respectively. In this case, the models include the batch normalization and the dropout layers in the output part of each hidden layer to prevent overfitting and improve normalization performance. The data preprocessing was applied in the same way as the CNN. As a result of the ATxmega128 instruction classification, the recovery accuracy of the validation data is 76.2% and the cross-board data accuracy is 70.6%. In addition, in the case of the STM32F0, we confirmed that the validation data can be recovered with an accuracy of 98.2% and cross-board data of 93.6%. Since this shows the best results in the cross-board data, we tried to recover by improving the MLP model.

The improved MLP model has 10 hidden layers, and each hidden layer consists of 512 neurons. In addition, batch normalization and dropout were applied as in the simple MLP, and a residual block was added. As a result of the ATxmega128 instruction classification, the recovery accuracy of the validation data is 91.9% and the cross-board data accuracy is 77.0%. In addition, in the case of the STM32F0, we confirmed that the validation data can be recovered with an accuracy of 98.6% and cross-board data of 96.5%. 

The following Figure 8 shows the confusion matrices of the single board and cross-board for the ATMEGA128 and STM32F0, respectively, when using an MLP-10 model. The x-axis of the confusion matrix is the predicted instruction by the model, and the y-axis is the actual instruction. The confusion matrix can provide insight by showing the accuracy of the model for each label.

In addition, we show the instruction recovery accuracy of the ATxmega128 and the STM32F0 when using all models in Table 3. It is not yet clear why the complex 32-bit instructions are more distinguishable than the 8-bit instructions. Nevertheless, we can infer that this is because the number of 32-bit instructions selected for the experiment is less than the number of 8-bit instructions. 

The SCBD can be utilized in profiling scenarios rather than non-profiling scenarios. In other words, it is necessary to check whether the template created by the profiling device is valid in the target device. Therefore, we focused on cross-board accuracy to test the SCBD. The following Table 4 shows a comparison of our SCBD with the previous research in terms of cross-board accuracy. We confirmed that our SCBD can distinguish instructions of the 32-bit Cortex-M0 with an accuracy of about 96.5%, whereas all other previous studies did not evaluate the cross-board accuracy.

## 6. Conclusions

Side-channel analysis is mainly studied in the field of cryptanalysis to extract the secret key of cryptographic algorithms. However, several studies have shown that the instructions executed inside the microcontroller can also be recovered using side-channel signals such as power consumption and electromagnetic radiation. In this paper, we implemented the SCBD for the 8-bit ATxmega128 and the 32-bit STM32F0, which are commercial microcontrollers, using the side-channel power consumption leakage. We confirmed that instructions can be recovered with 91.9% and 98.6% accuracy in the ATxmega128 and the STM32F0, respectively, using deep learning models to which various normalization techniques were applied. Furthermore, we achieved an accuracy of 77% and 96.5% in the ATxmega128 and STM32F0, respectively, in cross-board validation. Actually, the power consumption of the microcontroller includes signals from not only the processor, but also peripheral devices such as the flash memory and SRAM. In other words, there is much switching noise, which is a signal that is not related to the instruction. Nevertheless, we confirmed that the instruction can be recovered with high accuracy using the power consumption signal. Therefore, it will be possible to develop a more sophisticated disassembler that can completely recover the operands and literal constants if the local side-channel leakages are used and bit-level access is combined. However, the SCBD still has several limitations, such as accuracy problems and changes in the experimental environment. For this reason, more research about the SCBD is needed to apply to the real world.

## Figures and Tables

**Figure 1 sensors-22-05900-f001:**
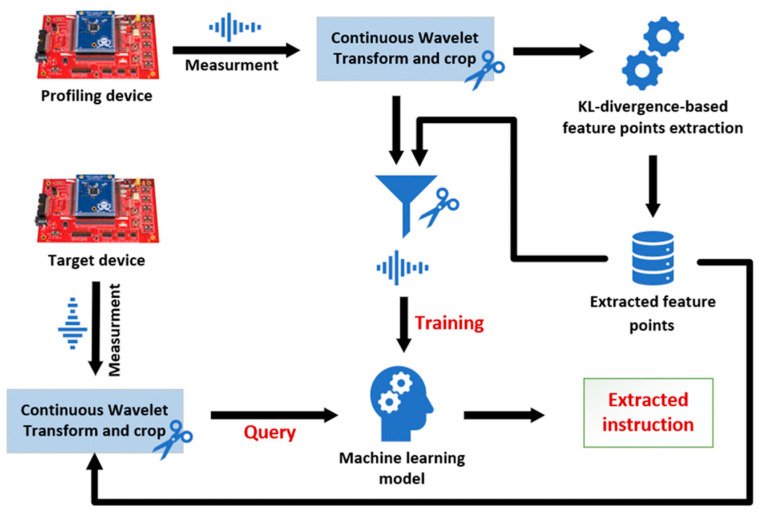
The overall flow of the side-channel-based disassembler.

**Figure 2 sensors-22-05900-f002:**
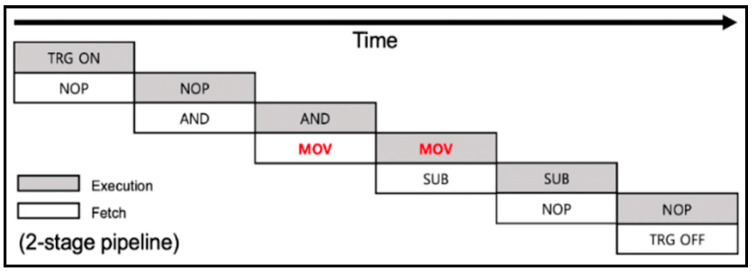
Single-level pipeline applied to the ATxmega128.

**Figure 3 sensors-22-05900-f003:**
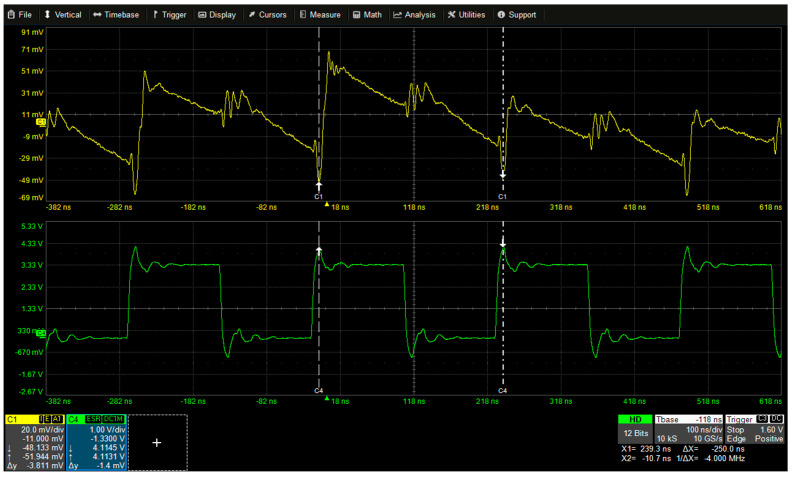
The power consumption pattern of the ATxmega128 (4 MHz).

**Figure 4 sensors-22-05900-f004:**
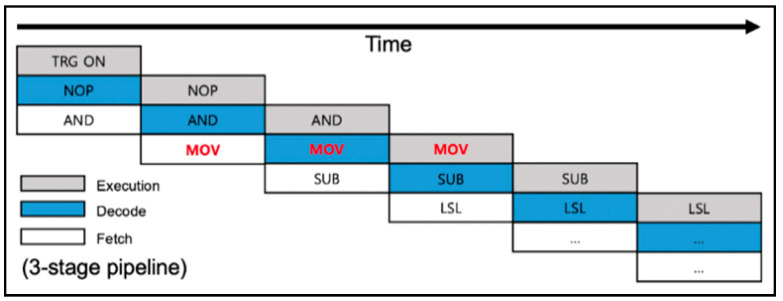
Three-stage pipeline applied to the Cortex-M0.

**Figure 5 sensors-22-05900-f005:**
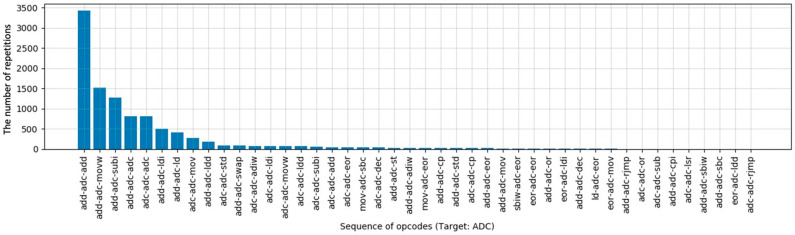
The result of instruction sequence analysis (ADC of the ATxmega128).

**Figure 6 sensors-22-05900-f006:**
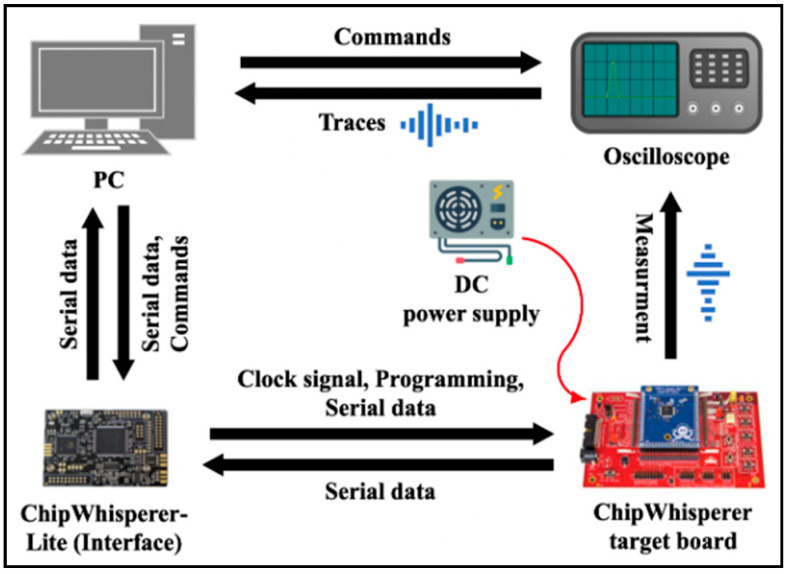
Instruction template acquisition structure using an oscilloscope and ChipWhisperer platforms.

**Figure 7 sensors-22-05900-f007:**
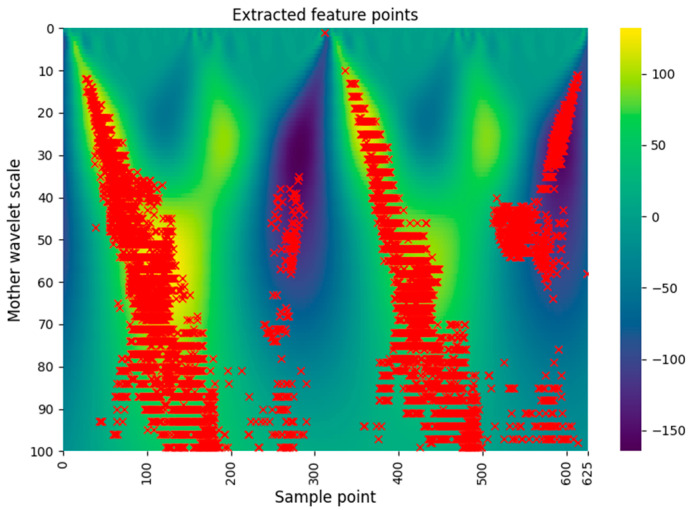
Final feature points of the ATxmega128.

**Figure 8 sensors-22-05900-f008:**
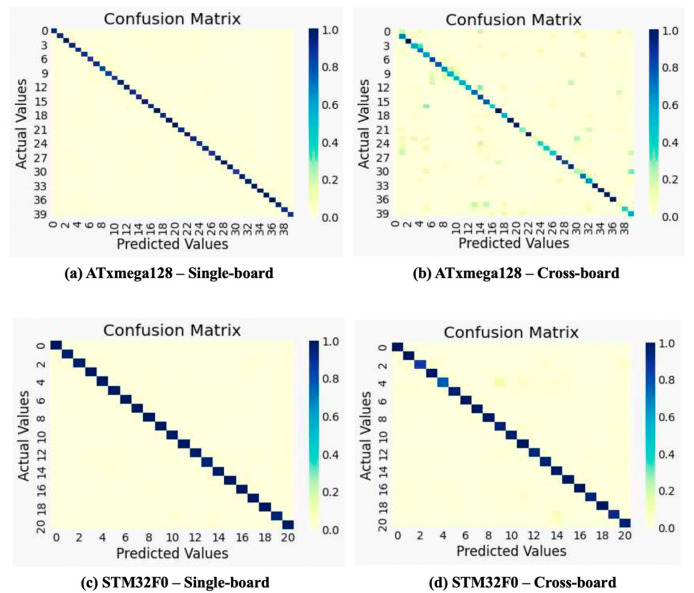
Confusion matrices of single-board and cross-board instruction recovery using MLP-10.

**Table 1 sensors-22-05900-t001:** Target opcodes of the ATxmega128 and the STM32F0 (Cortex-M0).

Target Opcodes of the ATxmega128					
ADD	ADC	ADIW	SUB	SUBI	SBC
SBCI	SBIW	AND	ANDI	OR	ORI
EOR	SBR	INC	DEC	RJMP	CP
CPC	BREQ	BRNE	BRCS	BRCC	BRLT
BRGE	MOV	MOVW	LDI	LDS	LD
LDD	STS	ST	STD	IN	OUT
ASR	SWAP	LSR	ROR		
**Target Opcodes of the STM32F0 (16-Bit Thumb of Cortex-M0)**					
ADC	ADD	AND	ASR	BIC	EOR
LDR	LDRH	LDRB	LSL	LSR	MOV
ORR	REV	ROR	STRH	STRB	SUB
SXTB	UXTB	UXTH			

**Table 2 sensors-22-05900-t002:** The number of measured power consumption traces according to the microcontroller.

	ATxmega128		STM32F0	
	Training/Validation	Cross-Board	Training/Validation	Cross-Board
# of opcodes	40	40	21	21
# of firmware per opcode	30	5	10~25	5
# of traces per firmware	100	25	100	25
Total	120,000	5000	36,000	2625

**Table 3 sensors-22-05900-t003:** Instruction recovery accuracy using machine learning and deep learning models.

	ATxmega128		STM32F0	
	Validation	Cross-Board	Validation	Cross-Board
KNN	90.8%	64.2%	98.8%	84.6%
RF	91.3%	59.7%	98.7%	81.6%
CNN	66.7%	54.4%	84.8%	74.2%
MLP (3-layer)	76.2%	70.6%	98.2%	93.6%
MLP (10-layer)	91.9%	77.0%	98.6%	96.5%

**Table 4 sensors-22-05900-t004:** The comparison between our side-channel-based disassembler and existing disassemblers.

	Microcontroller	# of Instructions	Single-Board Validation	Cross-Board Validation
Ref. [14]	PIC16F687	33	70.01%	Not evaluated
Ref. [15]	PIC16F687	33	96.24%	Not evaluated
Ref. [16]	ATmega328P	112	99.03%	Not evaluated
Ref. [24]	LPC1768(32-bit)	61	98.0%	Not evaluated
Ref. [25]	ATmega8A	54	98.21%	Not evaluated
Our	ATxmega128	40 (77)	91.9%	77.0%
Our	STM32F0 (32-bit)	21	98.6%	96.5%

## Data Availability

Not applicable.

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
