# Peer review of "Implementation of Disassembler on Microcontroller Using Side-Channel Power Consumption Leakage"

_sensors, 2022, doi:10.3390/s22155900_

Round 1

Reviewer 1 Report

It feels a bit like the contribution of this paper is incremental. And mainly contributes in enlarging the body of knowledge and experimental results. Also a benefit is the wavelet based distinguisher with the KL feature extraction.

I do not see a really new perspective of showing something different than (e.g.) the Markov-chain paper let alone working better on the distinguisher and providing more results --> It seams the context should be changed to reflect that...

1/SCA countermeasures would generally defend such attacks as they are designed to protect P_{data} based information, how ever it is interesting to consider in a bit better detailed discussion their ability to defend such attacks. I am lacking a clear discussion on to what extent such countermeasures should be used to reduce such a threat and to what extent?

Perhaps better relate to recent (e.g.) software countermeasures such as  SW oriented masking [1] -- would it be efficient ? is it an "over-kill" ???:

[1] Salomon, D. et-al. (2022). On the Performance Gap of a Generic C Optimized Assembler and Wide Vector Extensions for Masked Software with an Ascon-{\it {p}} test case. Cryptology ePrint Archive.

Also to a more HW related protection layers - such as simple (and low-cost) power-randomization [2] - is it enough / what are your expectation ??:

[2] Breuer, et-al. (2022). Fully-Digital Randomization Based Side-Channel Security—Towards Ultra-Low Cost-per-Security. IEEE Access.

2/ Many claims a "rough", not substantiated and generally not correct and correctness +structure should be improved: "SPA based on Pop can be easily defended by using a constant 97 time operation, Pop is only used to find an overall structure of cryptographic algorithms" --> this is not correct. it is both the timing which leaks but also the data-dep instruction procedure. many countermeasure to have a constant-time but also constant instruction stream data independent (e.g., to defend horizontal/single trace attacks).

3/ I will not just list  HW/ HD models and that's it "per se", it is detached from the context -- rather I would say the leakage is ultra complex in devices and cannot be contained by a simple model. however in standard unprotected devices such simplified leakage models already correlates nicely and provide some level of information. However, in you "template" case you build your own model and just need to decide on labels and classes ... I would try to link it better.

4/ I would be happy to see not only "how" different instructions are tractable" but also --> what is the SNR, i.e. how data-dependent the pattern is and what is the noise level in this context (or the Signal level if you try to extract data dependency with your tool).

5/ please be careful this is a specific observation which is platform dependent and should not be discussed as something general "showed that electromagnetic radiation containing much more information"

6/ must be treated and better explained: Section 3.2.1 - you illustrate an in-line assembler code of 3 instructions in a 3 stage pipeline --> as long as either on optimizations allowed by the (say) gcc compiler or all optimizations -O3 and depending on the data-path parallelism/out-of-order/reordering buffer etc. --> Generally, you can restrict and template each and every combination separately... moreover, surely not all 137^3 options occur and you can specifically make sure to template all... This section is really not clear / not well justified and not consistent. please specifically layout your setting  / the different options and assumptions.

>> Relating to 3.2.2 >> this is clear. To me section 3.2.1 discussed a not realistic baseline discussion.

>> Why would the Markov-chain analysis paper would not see this relation / connection >> i.e. profile only the viable instructions-sequences really taking place in the HW ?

7/ generally in any FFT-based of Wavelet-based the accuracy of the model is dominated by the length of the (prior to transform) sequence and Shannon Sampling principle, if it is too short you loose information, since the sequence is short in #clock cycles I wonder what part of the information you do capture. I think this should be justified and discussed. You can also relate or add one ref. regarding the utilization of Wavelet transform in the SCA context.

8/ for feature-extraction and statistical- distance  using KL in the SCA context you can relate to some prior art e.g., [3]

in-line asm. figure would be nicer in a pseudo code or from an editor and not as a cut figure.

[3] R. Breuer and I. Levi, "How Bad Are Bad Templates Optimistic Design Stage Side-Channel Security Evaluation and its Cost"

9/ foldnum and divider should be well defined

10/ why only one dimensional CNN ?

Language can be improved, 

"The side-channel analysis " - there is no specific SCA it is a class/type/group, the is not needed.

Sentence structure "cause">"result". 

E.g., "Since" repeats again and again and again in one paragraph - not necessary wrong in this context but feels bad. 

SPA - not "the DPA"

"we analyze two ... that are being analyzed" - fix

Author Response

Response to Reviewer 1 – (Round 1)

  1. As you pointed out, we added additional explanations and references below.
  • Salomon, D. et al., On the Performance Gap of a Generic C Optimized Assembler and Wide Vector Extensions for Masked Software with an Ascon-p test case
  • Breuer, R. et al., Fully-Digital Randomization Based Side-Channel Security—Towards Ultra-Low Cost-per-Security

  1. We deleted the wrong parts and revised that part of the paper according to your comment. (Section 2.1)

  1. We added a description of the leakage model according to your advice. In addition, the association between the use of deep learning in our disassembler and the complexity of the leakage model is described in section 2.1.

  1. According to your comments, we partially added descriptions in sections 3~4.

  1. We removed the ambiguous sentences according to your point.

  1. According to your comments, we added some descriptions to Section 2.

  1. We do not know which frequency band contains the information in the power consumption signal in the clock cycle. We performed feature extraction by dividing the scale into 100 equal parts from the very low frequency to the high frequency. As a result, only meaningful features can be actually selected by the deep learning model.

  1. According to your comments, we modified figure 4(figure 5 in the revised paper) from a cut figure to a listing.

  1. According to your comments, we modified section 4.2.1. (The ‘foldnum’ represents the number of repetitions of feature extraction to improve reliability. And the divider is used to calculate indices of trace.)

  1. In general, the two-dimensional CNN performs training while maintaining association with neighboring data pixels of a picture. Since the data used in our experiment were extracted from the spectrogram and concatenated as shown in figure 9, there is no correlation between neighboring data. Considering this point, we adopt one-dimensional CNN.

Reviewer 2 Report

In general, the article makes a good impression. It can be accepted after the elimination of the following remarks:

1. P. 7. L. 260. A space before the parenthesis is needed.

2. Figure 5. Frequency units not specified. Frequency is usually measured in units per time interval. What interval? Rather, it is the number of repetitions of opcodes sequences.

3. The relevance of this study is not entirely clear. The fact is that the electrical parameters of electronic devices, even from the same batch, may differ slightly. Not to mention the fact that the ATxmega128 microcontroller can be produced by different fabrics and according to different technological processes. Where is the guarantee that the measured parameters on the authors' target boards will match any ATxmega128 and STM32F0?

4. The section with machine learning is insufficiently described. Why were such machine learning methods chosen? How did the training and validation take place? Why, not used, for example, XGBoost? After all, K-Nearest Neighbor and the Random Forest are quite old methods, and more advanced ones have already been proposed. There is no justification for choosing the number of layers and nodes in a multi-layer perceptron. It is not clear whether the models performed calculations in floating point or fixed point format (this is important for the possible implementation of the proposed disassembly method in the form of a hardware stand)?

5. It has not become clear to me from the article: why the instruction recovery accuracy for the much simpler ATxmega128 controller is much worse than for the STM32F0?

6. Is 91.9% accurate enough? How close will the program be restored? Does it make any practical sense at all? Indeed, often the distortion of even one instruction can completely destroy the logic of the program. And here it turns out that almost every 10th instruction is distorted.

Author Response

Response to Reviewer 2 – (Round 1)

  1. We revised P.7, L. 260 (P. 7, L. 281 in a revised paper) according to your comments.

  1. A "Frequency" of Figure 5 (Figure 6 in a revised paper) is just the number of repetitions of opcodes sequences. We modified Figure 5 (frequency -> The number of repetitions).

  1. As you pointed out, microcontrollers produced in different processes are likely to have different characteristics. Therefore, we removed any misleading parts of the paper. In addition, we clearly specified the experimental equipment.

  1. Our main purpose is to apply deep learning models. Therefore, we selected KNN, and RF, which are the most widely used algorithms in previous research for comparison with our model. In addition, most of the machine learning models, not the deep learning models, did not have a significant performance difference.

  1. Unfortunately, we couldn't give a clear reason for that. But the reason is presumed that the number of instructions selected in the STM32F0 experiment is smaller than ATxmega128. We described this reason in section 5.

  1. The side-channel-based disassembler is still insufficient to be applied in the real world. There may be various reasons for this, and the most representative ones are the accuracy problem and environmental change problem. Almost side-channel-based disassembler papers, including this one, are constantly looking for ways to improve.

Reviewer 3 Report

In this paper, the authors presented the Implementation of a disassembler on a microcontroller using side-channel power consumption leakage, and they have confirmed based on the experimental results that the operating instructions of ATxmega128 and STM32FO can be recovered with high accuracy through deep learning models. However, I observed that the paper has the following limitations:

11)     Generally, I feel the novelty of this paper is very limited since it only implements existing recovery techniques of the operating instructions. Furthermore, the efficiency of a disassembler should be compared with related works.

22)     Please describe your overall implementation (including recovery of operating instructions) flow using a flowchart/figure for easy-to-follow paper.

33)      The authors should refer to many related research papers and compare your work with them in terms of area, throughput, efficiency, power, etc.

44)      The authors can provide the key contributions of this paper as a separate subsection in the introduction.

55)      the authors can use a comparison table for showing the limitations of the previously proposed operating instructions recovery methods with your method.

66)      However, I observed that there are lots of disassembler methods that are not compared in this paper in terms of instructions recovery accuracy. The comparison is incomplete. I think that the authors should better frame their work by referring to and comparing their proposed implementation to a more complete and updated set of the published recent literature. Some relevant papers:

a. Eisenbarth, T.; Paar, C.; Weghenkel, B. Building a side channel-based disassembler. Transactions on Computational Science X, 2010, 6340, 536 pp. 78-99.

b. Strobel, D.; Bache, F.; Oswald, D.; Schellenberg, F.; Paar, C. Scandalee: a side-channel-based disassembler using local electromagnetic emanations. In Proceedings of DATE’15, Grenoble, France, 9-13 March 2015; pp. 139-144.

c. Park, J.; Xu, X.; Jin Y.; Forte, D. Power-based side-channel instruction-level disassembler. In Proceedings of DAC’18, San Francisco, CA, USA, 24-28 June 2018; pp. 1-6.

d. Park, J., Anandakumar, N.N., Saha, D., Mehta, D., Pundir, N., Rahman, F., Farahmandi, F. and Tehranipoor, M.M., 2022. PQC-SEP: Power Side-channel Evaluation Platform for Post-Quantum Cryptography Algorithms. IACR Cryptol. ePrint Arch., 2022, p.527.

e. P. Narimani, M. A. Akhaee and S. A. Habibi, "Side-Channel based Disassembler for AVR Micro-Controllers using Convolutional Neural Networks," 2021 18th International ISC Conference on Information Security and Cryptology (ISCISC), 2021, pp. 75-80, doi: 10.1109/ISCISC53448.2021.9720466.

77)     From an experimental point of view, the results presented in section 5 are not significant to support the authors' claims for recovery of the operating instructions. The authors should give in more experimental results to the recovery of operating instructions (for example you can use the resulting chart that will give more confidence to the reader).

88)     I think one of the biggest problems is writing because there are many grammar mistakes. The editorial quality of the paper is not up to the mark. The author(s) should have done a thorough proofread of the manuscript before submission. 

Author Response

Response to Reviewer 3 – (Round 1)

  1. We have included the contribution of our paper to section 1. And we added the comparison of some results of our paper with the existing research in section 5.

  1. We added a figure that can represent a flow of construction and utilization of the disassembler in section 1.

  1. According to your comments, we added the comparison of some results of our paper with the existing research in section 5.

  2. We have summarized the contributions of this research and added a subsection describing them in section 1.

  1. We added a comparison of some results of our paper with the existing research in section 5.

  1. We added a comparison of some results of our paper with the existing research in section 5. (Comparison table)

  2. According to your comments, we included a confusion matrix as an additional figure to the experimental results (in Section 5).

  3. We first proofread the paper and correct some grammatical errors. Due to the lack of proofreading time, we will continue to improve after the submission of this revision.

Round 2

Reviewer 1 Report

  1. You have added literature on DPA countermeasures, thanks. However, you completely missed the point: DPA countermeasures can conceptually protect both P_{data} and P_{OP} depending on how they are designed ... The point here was to state that conceptually DPA can potentially protect such breaches.  

  1. I didn't catch exactly where you have referred to this: "SPA based on Pop can be easily defended by using a constant time operation, Pop is only used to find an overall structure of cryptographic algorithms" --> this is not correct. it is both the timing which leaks but also the data-dep instruction procedure. many countermeasure to have a constant-time but also constant instruction stream data independent (e.g., to defend horizontal/single trace attacks)."

  1. "According to your comments, we added some descriptions to Section 2." I AM REPOSTING THIS I did not see a clear discussion and reply: must be treated and better explained: Section 3.2.1 - you illustrate an in-line assembler code of 3 instructions in a 3 stage pipeline --> as long as either on optimizations allowed by the (say) gcc compiler or all optimizations -O3 and depending on the data-path parallelism/out-of-order/reordering buffer etc. --> Generally, you can restrict and template each and every combination separately... moreover, surely not all 137^3 options occur and you can specifically make sure to template all... This section is really not clear / not well justified and not consistent. please specifically layout your setting  / the different options and assumptions.

    >> Relating to 3.2.2 >> this is clear. To me section 3.2.1 discussed a not realistic baseline discussion.

    >> Why would the Markov-chain analysis paper would not see this relation / connection >> i.e. profile only the viable instructions-sequences really taking place in the HW ?

  2. Where do you state this inline ?

  1. In general, the two-dimensional CNN performs training while maintaining association with neighboring data pixels of a picture. Since the data used in our experiment were extracted from the spectrogram and concatenated as shown in figure 9, there is no correlation between neighboring data. Considering this point, we adopt one-dimensional CNN.>>> Is this well explained in the text ?

Author Response

Response to Reviewer 1 – (Round 2)

  1. We understood your comment well. We added an additional sentence in section 2.1(L 119 - L 121) according to your advice.

è These countermeasures based on their specific design protect not only but also  from the advanced side-channel analysis attacks including DPA, CPA, etc.

  1. We recognized that the sentence you mentioned was incorrect and deleted it. And we added to L 115 that single-trace attacks using Pop are possible.

è In the power analysis attack targeting , timing information or data-dependent information can be utilized in SPA, single trace attacks (using correlation coefficient), and so on.

  1. We removed sentences that could cause misunderstanding and added content related to HMM (L 255 – L 259).

è (Section 3.2 – Removed) Since the ATxmega128 supports 137 opcodes, the number of combinations of opcodes that can come out of the configuration shown in Fig. 5 is 137^3 (about 2.5 million). If an instruction template is constructed by selecting the opcodes randomly among them, only a small part of the 2.5 million can be composed as an instruction template. However, in the randomly selected templates, combinations that the compiler cannot generate will occupy most of them. Furthermore, there are even opcodes that are not used at all.

è (Section 3.2 - Added) Therefore, only instruction sequences that can actually be executed in hardware should be constructed as templates. To select instructions that can actually operate on the hardware, we compile and analyze real block cipher algorithms instead of using the Hidden Markov Model (HMM). This method can be simply used when extracting instructions as an alternative to the complex HMM model.

  1. We added an additional explanation in section 4.1.2 (L 371 – L 376) according to your advice.

è In our experiment environment,  is 2 because a signal of 2 clocks is used,  is 0.25, is Hz, and ∆ is , so the maximum scale s is determined to be 156.25. In fact, we do not know which frequency band contains meaningful information. Therefore, we use the value obtained by dividing the space between 0 and 156.25 into 100 equal parts as the scale. As a result, only meaningful features can be automatically selected by the deep learning model.

  1. We added a description of CNN in section 5.1.3 2 (L 458 – L 463) according to your advice.

è To classify instructions using CNN, we selected one-dimensional CNN rather than two-dimensional CNN. In general, the two-dimensional CNN performs training while maintaining association with neighboring data such as pixels of a picture. Since the data used in our experiment were extracted from the spectrogram and concatenated as shown in Fig. 10, there is no correlation between neighboring data. Considering this point, we adopt one-dimensional CNN in this experiment.

Reviewer 2 Report

In general, I am satisfied with the corrections and the authors' answers. I think that the article can be published.

It is necessary in the conclusions to focus on the main result related to cross-board accuracy. And also there should be a reflection on the accuracy, how much it is sufficient to apply the described solution in real life.

Author Response

Response to Reviewer 2 – (Round 2)

  1. Thank you for your insightful review and helpful comments.
    We added a cross-board accuracy and some opinions about real-world applications in conclusions.

Reviewer 3 Report

The authors have addressed my previous concerns. I support acceptance. However, the authors may include all references as I suggested in my previous review comments all relevant to side-channel power analysis.

Author Response

Response to Reviewer 3 – (Round 2)

  1. Thank you for your insightful review and helpful comments.

We have added the four references you suggested
